# A positive feedback loop between ZEB2 and ACSL4 regulates lipid metabolism to promote breast cancer metastasis

Jiamin Lin[1†], Pingping Zhang[1†], Wei Liu[2†], Guorong Liu[1], Juan Zhang[1], Min Yan[3*], Yuyou Duan[4*], Na Yang[1*]

[1]Department of Laboratory Medicine, The Second Affiliated Hospital, School of Medicine, South China University of Technology, Guangzhou, China; [2]Department of Breast Surgery, Guangzhou Red Cross Hospital, Guangzhou Red Cross Hospital of Jinan University, Guangzhou, China; [3]Sun Yat-sen University Cancer Center, State Key Laboratory of Oncology in South China, Collaborative Innovation Center of Cancer Medicine, Guangzhou, China; [4]Laboratory of Stem Cells and Translational Medicine, Institutes for Life Sciences and School of Medicine, South China University of Technology, Guangzhou, China

**\*For correspondence:**
yanmin@sysucc.org.cn (MY);
yuyouduan@scut.edu.cn (YD);
eyyangna@scut.edu.cn (NY)

[†]These authors contributed equally to this work

**Abstract** Lipid metabolism plays a critical role in cancer metastasis. However, the mechanisms through which metastatic genes regulate lipid metabolism remain unclear. Here, we describe a new oncogenic–metabolic feedback loop between the epithelial–mesenchymal transition transcription factor ZEB2 and the key lipid enzyme ACSL4 (long-chain acyl-CoA synthetase 4), resulting in enhanced cellular lipid storage and fatty acid oxidation (FAO) to drive breast cancer metastasis. Functionally, depletion of ZEB2 or ACSL4 significantly reduced lipid droplets (LDs) abundance and cell migration. ACSL4 overexpression rescued the invasive capabilities of the ZEB2 knockdown cells, suggesting that ACSL4 is crucial for ZEB2-mediated metastasis. Mechanistically, ZEB2-activated ACSL4 expression by directly binding to the ACSL4 promoter. ACSL4 binds to and stabilizes ZEB2 by reducing ZEB2 ubiquitination. Notably, ACSL4 not only promotes the intracellular lipogenesis and LDs accumulation but also enhances FAO and adenosine triphosphate production by upregulating the FAO rate-limiting enzyme CPT1A (carnitine palmitoyltransferase 1 isoform A). Finally, we demonstrated that ACSL4 knockdown significantly reduced metastatic lung nodes in vivo. In conclusion, we reveal a novel positive regulatory loop between ZEB2 and ACSL4, which promotes LDs storage to meet the energy needs of breast cancer metastasis, and identify the ZEB2–ACSL4 signaling axis as an attractive therapeutic target for overcoming breast cancer metastasis.

## eLife assessment

This study provides a **valuable** finding on the mechanistic connections between epithelial-mesenchymal transition (EMT) and lipid metabolism. The authors identified the ZEB2/ACSL4 axis as a newly discovered metastatic metabolic pathway that promotes both lipogenesis and fatty acid oxidation. The evidence supporting the claims of the authors is **solid**. The work will be of interest to medical biologists working on cancer.

## Introduction

More than 90% of breast cancer-related deaths are due to metastasis (*Fahad Ullah, 2019*; *Park et al., 2022*). Current treatments, including endocrine therapy, chemotherapy, and radiation therapy,

are ineffective in preventing breast cancer metastasis and remain the greatest clinical challenge for breast cancer treatment (*Barzaman et al., 2020*). Cancer metastasis is associated with a process called epithelial–mesenchymal transition (EMT) (*Mittal, 2018*; *Lüönd et al., 2021*). EMT is a pre-metastatic state in which epithelial cells lose their tight junctions and convert to migratory mesenchymal cells (*Dongre and Weinberg, 2019*). Metastatic invasion is a highly energy-intensive process (*Mosier et al., 2021*). It has become increasingly recognized that metabolic reprogramming during the EMT process contributes to metastasis and tumorigenesis (*Kang et al., 2019*; *Sciacovelli and Frezza, 2017*). Emerging evidence suggests that lipid metabolic reprogramming plays a critical role in meeting the energy requirements of metastatic invasion (*Bian et al., 2021*; *Li et al., 2020*; *Snaeb-jornsson et al., 2020*; *Cheng et al., 2018*). Elucidating the mechanism by which reprogrammed lipid metabolism helps us exploit novel and attractive targets for metastatic therapeutic interventions.

Lipid metabolism includes a complex network of pathways that regulate fatty acids (FAs) synthesis, storage, and degradation (*Maan et al., 2018*). For lipid anabolism, FAs are stored in a dynamic organelle called a lipid droplet (LD), which is composed of a monolayer of phospholipids that covers a hydrophobic core containing neutral lipids, such as triacylglycerol (TAG) and cholesterol esterase (CE) (*Cruz et al., 2020*). lipid droplets (LDs) accumulation is associated with aggressiveness in many cancer types (*Antunes et al., 2022*), including breast (*Zembroski et al., 2021*), brain (*Taïb et al., 2019*), liver (*Niu et al., 2021*), lung (*Jin and Yuan, 2020*), and prostate (*Roman et al., 2020*; *Huang et al., 2012*). Indeed, the aggregation of FAs in LDs is considered a priming state to prepare for metastasis (*Petan, 2020*; *Rozeveld et al., 2020*; *Senga et al., 2018*). In the case of need, FAs can be released and oxidized for energy support. For example, it has been reported that FAs stored in LDs were a crucial resource in fueling the metastatic process in pancreatic cancer (*Rozeveld et al., 2020*). Additionally, previous studies have suggested that metastatic triple-negative breast cancer (TNBC) depends on FAO to produce high adenosine triphosphate (ATP) levels (*Camarda et al., 2016*; *Wright et al., 2017*). Although lipid metabolism is crucial for cancer metastasis, the signaling pathway that regulates lipid metabolic reprogramming during breast cancer metastasis remains unclear.

Dysregulation of lipid metabolic enzymes has been documented in cancer metastasis (*Luo et al., 2017*). Long-chain fatty acyl synthetase 4 (ACSL4) belongs to the long-chain acyl-CoA synthetase ligase enzyme family (*Rossi Sebastiano and Konstantinidou, 2019*; *Quan et al., 2021*). ACSL4 catalyzes the conversion of long-chain FAs to acyl-CoAs, a necessary step for free long-chain FAs to enter the next metabolic pathway (*Kuwata and Hara, 2019*). The increased expression and activity of ACSL4 have been observed in many cancer types and it is well-known biomarkers of ferroptosis (*Doll et al., 2017*). Although it has been reported that ACSL4 is a tumor suppressor that activates ferroptosis, many studies have suggested that ACSL4 is an oncogene that contributes to tumor progression. For example, ACSL4 promotes hepatocellular carcinoma (HCC) cell proliferation and metastasis via lipogenesis and LDs accumulation (*Niu et al., 2021*; *Chen et al., 2020*). In prostate cancer, ACSL4 promotes cell growth, invasion, and hormonal resistance (*Wu et al., 2015*). The function of ACSL4 in breast cancer has been implicated in hormone therapy resistance involving the regulation of energy-dependent transporter expression (*Orlando et al., 2019*). However, regulation of lipid metabolism by ACSL4 during breast cancer invasion remains unclear.

In the present study, we demonstrate a novel positive feedback loop between the EMT transcription factor ZEB2 and the essential lipid metabolic enzyme ACSL4, resulting in enhanced cellular LDs accumulation and fatty acid oxidation (FAO) to drive breast cancer metastasis. Mechanistically, ZEB2 activates ACSL4 expression by directly binding to the ACSL4 promoter. ACSL4 stabilizes and upregulates ZEB2 via transcriptional and post-transcriptional mechanisms. In addition, we also provide evidence that overexpression of ZEB2 or ACSL4 is associated with worse prognosis in advanced breast cancer. Our findings provide insights into lipid metabolic mechanisms during the EMT process and reveal a novel oncogenic–metabolic pathway critical for breast cancer EMT and metastasis.

## Results
### ZEB2 and ACSL4 are overexpressed and correlated in highly invasive breast cancer cells

To explore the molecular mechanism of highly invasive breast cancer, we performed RNA-sequencing analysis of wild-type and two drug-resistant luminal breast cancer cell lines. Significant changes in 6155

genes were found (p < 0.05). Notably, EMT and stemness genes such as ZEB2, SNAIL, TWIST, Gli2, WNT, and AKT3, which are overexpressed in basal-like breast cancer (BLBC), were significantly upregulated in drug-resistant cells (*Figure 1—figure supplement 1A*). In contrast, differentiated genes such as FOXA1, ERα, E-cadherin, and GATA3, which are highly expressed in the luminal subtype, were significantly downregulated (*Figure 1—figure supplement 1B*), suggesting that drug-resistant cells underwent EMT and became stem-like cells. We noticed that the rate-limiting enzymes of FA metabolism, long-chain fatty acyl synthetase 4 (ACSL4), and EMT transcription factor ZEB2 were among the top 200 upregulated (>twofold) genes (*Figure 1A*, *Figure 1—figure supplement 2*). To verify these findings in the clinical sample, we analyzed ZEB2 and ACSL4 expression in TCGA database and found that ACSL4 expression was positively correlated with ZEB2 expression (*Figure 1B*, r = 0.7657, p < 0.001) and inversely correlated with ERα expression (*Figure 1C*, r = −0.3312, p < 0.001). by using the TCGA database, we compared the overall survival between ACSL4 or ZEB2 high- and low-expression breast cancer patients. We found that patients with higher ACSL4 or ZEB2 expression, especially those with simultaneous high expression had worse prognosis than those with lower expression (*Figure 1D–F*).

To confirm this correlation, we performed western blot analysis and found that basal-like and taxol-resistant cell lines, which lack ERα expression, showed high expression of ZEB2 and ACSL4, whereas low or no ZEB2 and ACSL4 expression was detected in luminal subtype cell lines (*Figure 1G*). Consistently, tissue samples from 45 breast cancer patients had similar expression patterns, with ZEB2 and ACSL4 being relatively highly expressed in ER-negative patient samples (patients 6, 7, 8, and 9) (*Figure 1—figure supplement 3A, B*). The immunohistochemistry staining assay, as shown in *Figure 1H–J*, also confirmed that the expression of ACSL4 was positively correlated with ZEB2 expression (*Figure 1H*, p < 0.001) and inversely correlated with ERα expression (*Figure 1I*, p < 0.001). Taken together, these results indicate that ACSL4 and ZEB2 are correlated and overexpressed in highly invasive breast cancers.

## Overexpression of ACSL4 contributes to ZEB2-mediated breast cancer invasion

Because ACSL4 is overexpressed in highly invasive breast cancer cells, we hypothesized that ACSL4 is essential for driving breast cancer migration and invasion. We then established a stable ACSL4 overexpressed MCF-7 cell line, which was less aggressive than the BLBC cells. The overexpression of ACSL4 significantly enhanced the metastatic and invasive capacities of MCF-7 cells (*Figure 2A, B*). Conversely, ACSL4 knockdown by shRNA significantly reduced the metastatic and invasive capacities of MDA-MB-231 cells compared to that of the control cells (*Figure 2C, D*). Interestingly, the Phalloidin staining showed that the ACSL4 knockdown cells had a significantly smaller length to width ratio, which indicates the reversion of EMT process, than those of the control group (p < 0.05) (*Figure 2—figure supplement 1*). We also observed that overexpression of ZEB2 significantly enhanced the metastatic and invasive capacities of MCF-7 cells (*Figure 2—figure supplement 2A, B*). Conversely, the metastatic and invasive abilities of MDA-MB-231 cells were significantly reduced in ZEB2 knockdown cells (*Figure 2—figure supplement 2C, D*). To investigate the downstream genes regulated by ACSL4, we performed RNA-sequencing analysis and identified that the tight junction and focal adhesion pathways were among the most upregulated pathways in ACSL4 knockdown cells compared to control cells (*Figure 2E*). To verify the RNA-seq results, we examined the expression of adhesion-related genes by immunoblotting and found that after silencing of ACSL4 and ZEB2, the luminal epithelial marker E-cadherin was increased, whereas the mesenchymal markers, such as vimentin and N-cadherin were decreased (*Figure 2F*). These results confirm the essential role of ACSL4 in breast cancer invasion and migration.

We hypothesized that the ZEB2–ACSL4 axis played a crucial role in metastatic ability. To examine whether ACSL4 was required for ZEB2-mediated breast cancer invasion and migration, ACSL4 was overexpressed in ZEB2-silencing cells (*Figure 2G, H*). As expected, overexpression of ACSL4 significantly restored the invasive and metastatic capacities of ZEB2 knockdown cells by 35.1% and 32.4%, respectively (*Figure 2I, J*), indicating that ACSL4 is essential for ZEB2-mediated breast cancer invasion and migration.

## ACSL4 and ZEB2 promote LDs production and lipogenesis

ACSL4 is a member of the long-chain acyl-CoA synthetase family, which catalyzes the conversion of long-chain FAs into their active forms. However, the mechanism by which ACSL4 regulates lipid

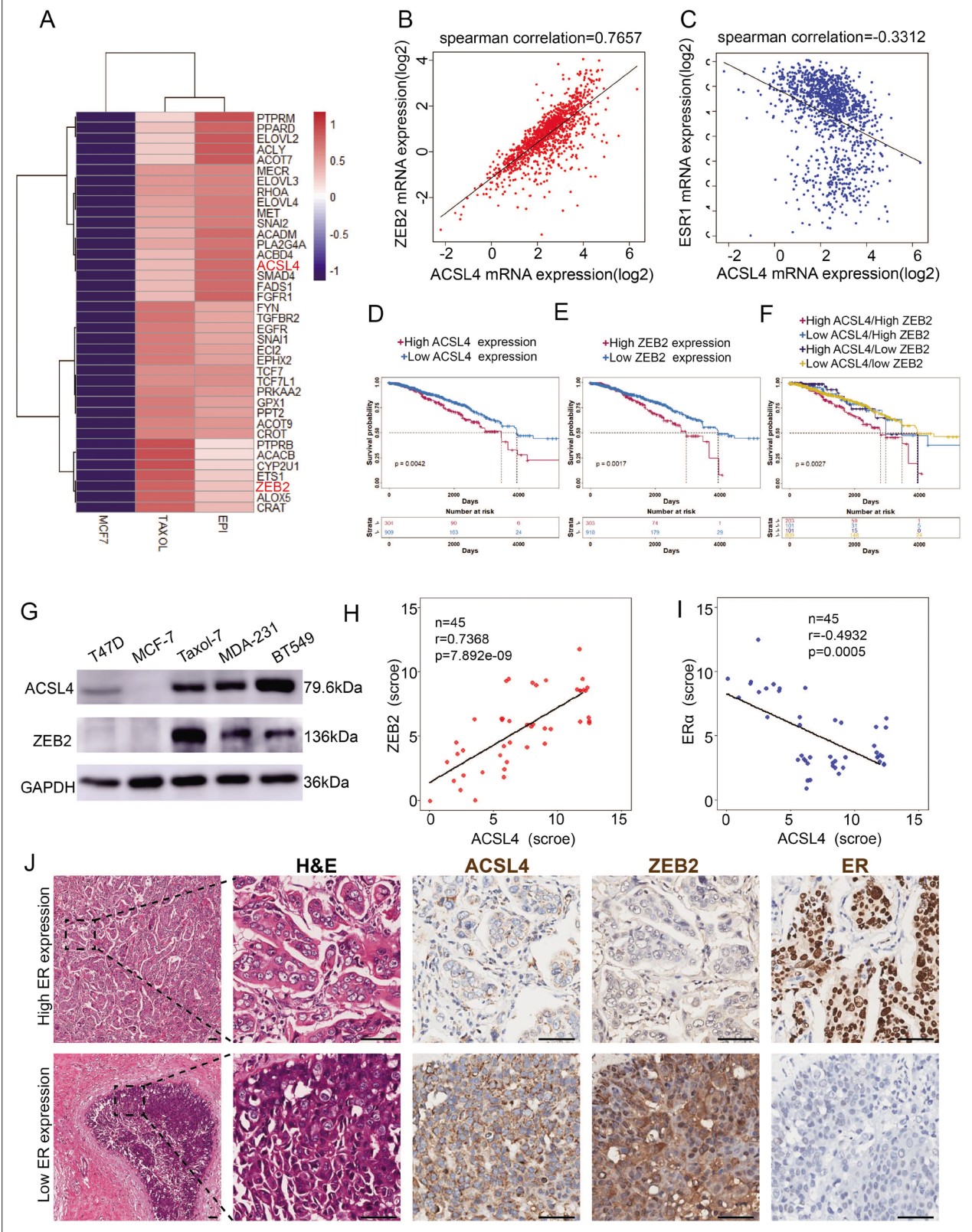

**Figure 1.** The expression and relationship of ZEB2 and ACSL4 in breast cancer. (**A**) Heatmaps of the 38 upregulated epithelial–mesenchymal transition (EMT)-related genes detected by RNA-seq analysis in the paclitaxel-resistant MCF-7 cell line (TAXOL) and epirubicin-resistant MCF-7 cell line (EPI) compared to wild-type MCF-7 cell line. (**B**) The correlation between ACSL4 and ZEB2 mRNA expression in the TCGA cohort consisting of 1222 breast cancer patient samples. Spearman correlation and linear regression analysis were employed. (**C**) The correlation between ACSL4 and ERα mRNA

*Figure 1 continued on next page*

*Figure 1 continued*

expression in the TCGA cohort consisting of 1222 breast cancer patient samples. Spearman correlation and linear regression analysis were employed. (**D**) OS (overall survival) was examined by Kaplan–Meier analysis to compare the survival rates in ACSL high and low expression of breast cancer patients. (**E**) OS (overall-progression survival) as examined by Kaplan–Meier analysis to compare the survival rates in ZEB2 high and low expression of breast cancer patients. (**F**) OS (overall-progression survival) was examined by Kaplan–Meier analysis to compare the survival rates in the four groups of breast cancer patients. (**G**) Expression of ACSL4 and ZEB2 was analyzed by western blot in a panel of five breast cancer cell lines, including two basal-like (MAD-231, BT549), two luminal (T47D, MCF-7), and a Taxol-resistant cell lines. (**H**) The correlation between ACSL4 and ZEB2 protein expression in the immunohistochemistry (IHC) cohort consisting of 45 breast cancer patient samples. (**I**) The correlation between ACSL4 and ERα protein expression in the IHC cohort consisting of 45 breast cancer patient samples. (**J**) HE staining and IHC analysis of ACSL4, ZEB2, ERα expression in representative basal-like and luminal subtype breast cancer tissues. Representative pictures were shown. Scale bar, 50 μm.

The online version of this article includes the following source data and figure supplement(s) for figure 1:

**Source data 1.** The gene expression datasets of the RNA-seq analysis of MCF-7 and EPI-resistant luminal breast cancer cell lines.

**Source data 2.** The gene expression datasets of the RNA-seq analysis of MCF-7 and Taxol-resistant luminal breast cancer cell lines.

**Source data 3.** The raw unedited gels or blots images of *Figure 1*.

**Figure supplement 1.** The representative differential genes between wild-type MCF-7 cells, paclitaxel-resistant MCF-7 cells (TAXOL), and epirubicin-resistant MCF-7 cells (EPI).

**Figure supplement 2.** The volcano plot was generated using R4.3.0 software for differentially expressed genes in the paclitaxel-resistant MCF-7 cell line (TAXOL) and epirubicin-resistant MCF-7 cell line (EPI) compared to wild-type MCF-7 cell line.

**Figure supplement 3.** The protein expression of ACSL4 and ZEB2 in patients with breast cancer.

metabolism in breast cancer remains unclear. A previous study revealed that ACSL4 promotes LDs accumulation in HCC (*Niu et al., 2021*). Thus, we measured the basal LDs content of breast cancer cells, including MCF-7, MDA-MB-231, and Taxol-resistant MCF-7 cells. The basal number and size of LDs were significantly larger in MDA-MB-231 and Taxol-resistant MCF-7 cells, both of which are highly invasive breast cancer cell lines (*Figure 3A*). Fluorescence microscopy revealed that ACSL4 co-localized with LDs in MDA-MB-231 cells (*Figure 3—figure supplement 1*). Notably, ACSL4 knockdown reduced cytoplasmic LDs abundance and LDs-containing cells in MDA-MB-231 cells (*Figure 3B*). Consistently, cytoplasmic LDs abundance and LDs-containing cells were significantly reduced in ZEB2-depleted cells (*Figure 3C*).

LDs are phospholipid monolayers containing a hydrophobic core comprising triacylglycerols (TG) and cholesterol esters (CE). As ACSL4 promotes intracellular LDs accumulation in breast cancer cells, we investigated the effect of ACSL4 on the lipid composition of breast cancer cells. Untargeted lipidomic analyses revealed that ACSL4 knockdown breast cancer cells had a reduced ability to incorporate long-strain monounsaturated (18:1, 17:1, 22:1, 24:1, and 26:1) and saturated FAs (16:0, 18:0, 24:0, and 26:0) into TG and phospholipids, indicating that ACSL4 promotes the incorporation of long-chain monounsaturated and saturated FAs into TG and phospholipids in these cells (*Figure 3D, E*). Moreover, all these lipids, including TG, phospholipids, and cholesterol esters, had decreased incorporation of several polyunsaturated FAs, such as 22:6, 20:4, 22:4, and 22:5, in ACSL4-depleted cells as reported previously (*Figure 3D*, *Figure 3—figure supplement 2A*; *Doll et al., 2017*). Consistently, the total amount of different lipid species, as shown in *Figure 3—figure supplement 2B*, were significantly decreased after ACSL4 knockdown. All together, these results suggest that ACSL4 directs the long train of free FAs into lipid anabolism to form different lipid species in the LDs or other cellular membranes.

## Exogenous FAs promotes LDs accumulation and fuel cell invasion

A previous study reported that some tumors tended to increase their intake of extracellular FAs to promote migration (*Rozeveld et al., 2020*). As we observed that depletion of ACSL4 greatly reduced cytoplasmic LDs abundance and invasive potential in BLBC cells, we hypothesized that LDs accumulation is an important step prior to breast cancer invasion. To verify our hypothesis, we examined whether exogenous FAs contributed to LDs accumulation and the invasive ability of breast cancer cells. Exogenous oleic acid (OA) was added to the cell culture medium. Treatment with OA dramatically enhanced LDs abundance in the cells (*Figure 4A*), indicating that breast cancer cells tend to store lipids in LDs for energy reserves. Next, we assessed the effects of exogenous OA treatment on cell migration. Transwell and wound healing assays revealed that OA-treated cells exhibited significantly

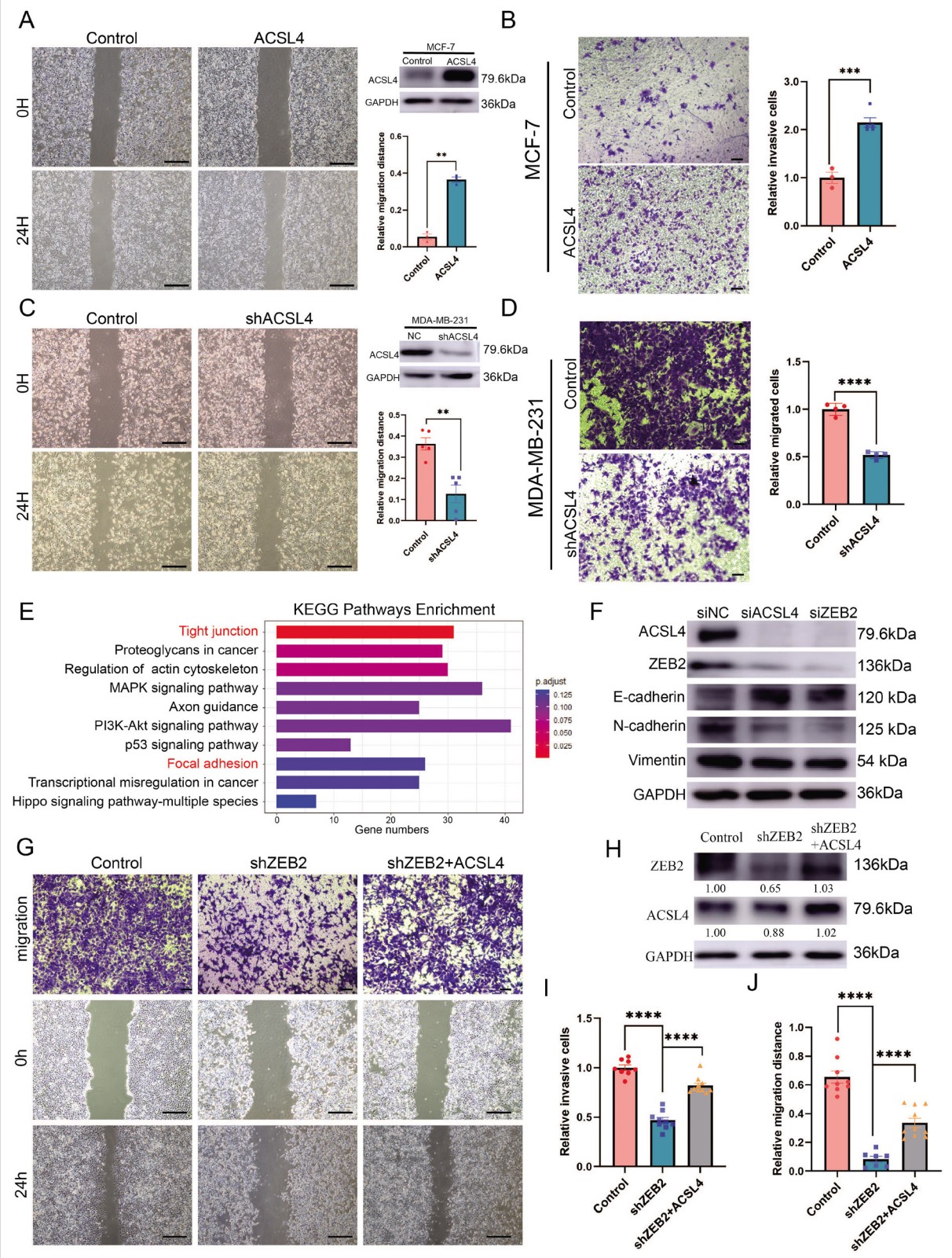

**Figure 2.** Overexpression of ACSL4 contributes to ZEB2-mediated breast cancer invasion. (**A**) Cell metastatic capacity was analyzed by wound healing assay in control or ACSL4 overexpression MCF-7 cells (left panel). Expression of ACSL4 was analyzed by western blot in control or ACSL4 overexpression MCF-7 cells. Quantification of relative migration distance (right panel). Scale bar, 5 mm. (**B**) Cell invasive capacity was analyzed by transwell invasion assay in control or ACSL4 overexpression MCF-7 cells (left panel). Quantification of relative invasive cells (right panel). Scale bar, 1 mm. (**C**) Cell

*Figure 2 continued on next page*

*Figure 2 continued*

metastatic capacity was analyzed by wound healing assays in control or ACSL4 silencing MDA-MD-231 cells (shACSL4) (left panel). Expression of ACSL4 was analyzed by western blot in control or ACSL4 silencing MDA-MD-231 cells (shACSL4). Quantification of relative migration distance (right panel). Scale bar, 5 mm. (**D**) Cell invasive capacity was analyzed by transwell invasion assay in control or ACSL4 silencing MDA-MD-231 cells (shACSL4) (left panel). Quantification of relative invasive cells (right panel). Scale bar, 1 mm. (**E**) KEGG (https://www.kegg.jp/) pathway enrichment analysis of differentially expressed genes by RNA-sequencing between control and ACSL4 knockdown MDA-MB-231 cells. The top 10 deferential pathways were listed. (**F**) Expression of three epithelial–mesenchymal transition (EMT)-related genes, E-cadherin, N-cadherin, and vimentin, was analyzed by western blotting in control, ACSL4, or ZEB2-silencing MDA-MB-231 cells. (**G**) Cell invasive and metastatic capacities were analyzed by transwell invasion assay and wound healing assays (scale bar, 1/5 mm) in control and ZEB2 knockdown, or ZEB2 knockdown cells that overexpress ACSL4. (**H**) Protein expression was analyzed by western blot in control, ZEB2 knockdown (shZEB2), and ZEB2 knockdown with ACSL4 overexpression (shZEB2 + ACSL4) cells. (**I**) Quantification of relative invasive cells in G. (**J**) Quantification of relative migration distance in G. Graphs indicated the statistical analysis in G analyzed by Student's *t*-test (mean ± standard error of the mean [SEM]). **p < 0.01, ***p < 0.001, ****p < 0.0001. All results are from three or four independent experiments.

The online version of this article includes the following source data and figure supplement(s) for figure 2:

**Source data 1.** The datasets of differentially expressed genes by RNA-sequencing between control and ACSL4 knockdown MDA-MB-231 cells.

**Source data 2.** The raw unedited gels or blots images of *Figure 2*.

**Figure supplement 1.** Fluorescence staining of Phalloidin in control or ACSL4 knockdown MDA-MB-231 cells.

**Figure supplement 2.** ZEB2 increases the metastatic and invasive capacities in breast cancer cells.

enhanced invasive and metastatic capacities compared with control cells (*Figure 4B, C*, *Figure 4—figure supplement 1*). To better determine the role of OA and ACSL4 on cell migration, the oleic acid (OA) was added in the culture medium of ACSL4 knockdown cells. As expected, the addition of oleic acid (OA) obviously restores the invasive and metastatic capacities of ACSL4 knockdown cells by 33.12% and 18.61%, respectively (*Figure 4D*).

Previous study reported that LDs undergo lipolysis during the process of migration in pancreatic cancer (*Rozeveld et al., 2020*). We reasoned that breast cancer cells utilize stored lipids during migration to fuel metastasis. LDs content was analyzed using fluorescence microscopy after cell migration. We observed that the lipid signal was significantly decreasing in the leading edge of the scratch of the wound healing migration (*Figure 4E*). In addition, we observed significantly reduced LDs in cells on the lower side of the transwell chamber, suggesting that lipids stored in LDs were utilized and degraded during cell migration and invasion (*Figure 4F*). Taken together, these results suggest that lipids stored in LDs are a crucial resource to fuel the process of breast cancer invasion and migration.

## ACSL4 and ZEB2 stimulate long-chain FAO and ATP generation in BLBC cells

To explore the mechanisms by which ACSL4 regulates lipid metabolism, we performed RNA-sequencing to investigate downstream pathways and genes regulated by ACSL4. KEGG enrichment analysis revealed that the FAO pathway was among the top 20 regulated pathways (*Figure 5A*). ACSL4 knockdown reduced the expression of genes involved in the FAO pathways (*Figure 5B*). Importantly, we observed that the FAO rate-limiting enzyme CPT1A was significantly reduced in ACSL4 knockdown cells (*Figure 5B*). Reverse transcription PCR (The polymerase chain reaction) confirmed that deletion of ACSL4 significantly reduced the expression of CPT1A, but did not affect the expression of CPT1B and CPT1C (*Figure 5C*). Consistently, ACSL4 silencing reduced CPT1A protein expression (*Figure 5—figure supplement 1A*). In addition, the levels of other lipid metabolic enzymes, such as ATGL, FASN, and SREBP2, were significantly decreased after ACSL4 knockdown (*Figure 5—figure supplement 1B*). As ACSL4 knockdown decreased CPT1A expression, we reasoned that ACSL4 might stimulate FAO in highly invasive breast cancer. Oxygen consumption rate (OCR) was calculated in ACSL4 or ZEB2-silencing and control cells. We observed that the OCR derived from long-chain FAs was significantly reduced in ACSL4 or ZEB2-silencing cells (*Figure 5D, E*), suggesting the metabolic advantage of increased long-chain FAO and oxidative phosphorylation (OXPHOS).

Since OXPHOS is accompanied by increased ATP generation, we measured ATP levels and found that ACSL4 or ZEB2 knockout cells had significantly reduced ATP generation compared to control cells (*Figure 5F, G*). To exclude OXPHOS derived from the aerobic glycolysis pathway, we measured lactate production and found no significant difference in lactate production between ACSL4 or ZEB2 knockdown and control cells, suggesting that glucose metabolism is not involved in ACSL4- or

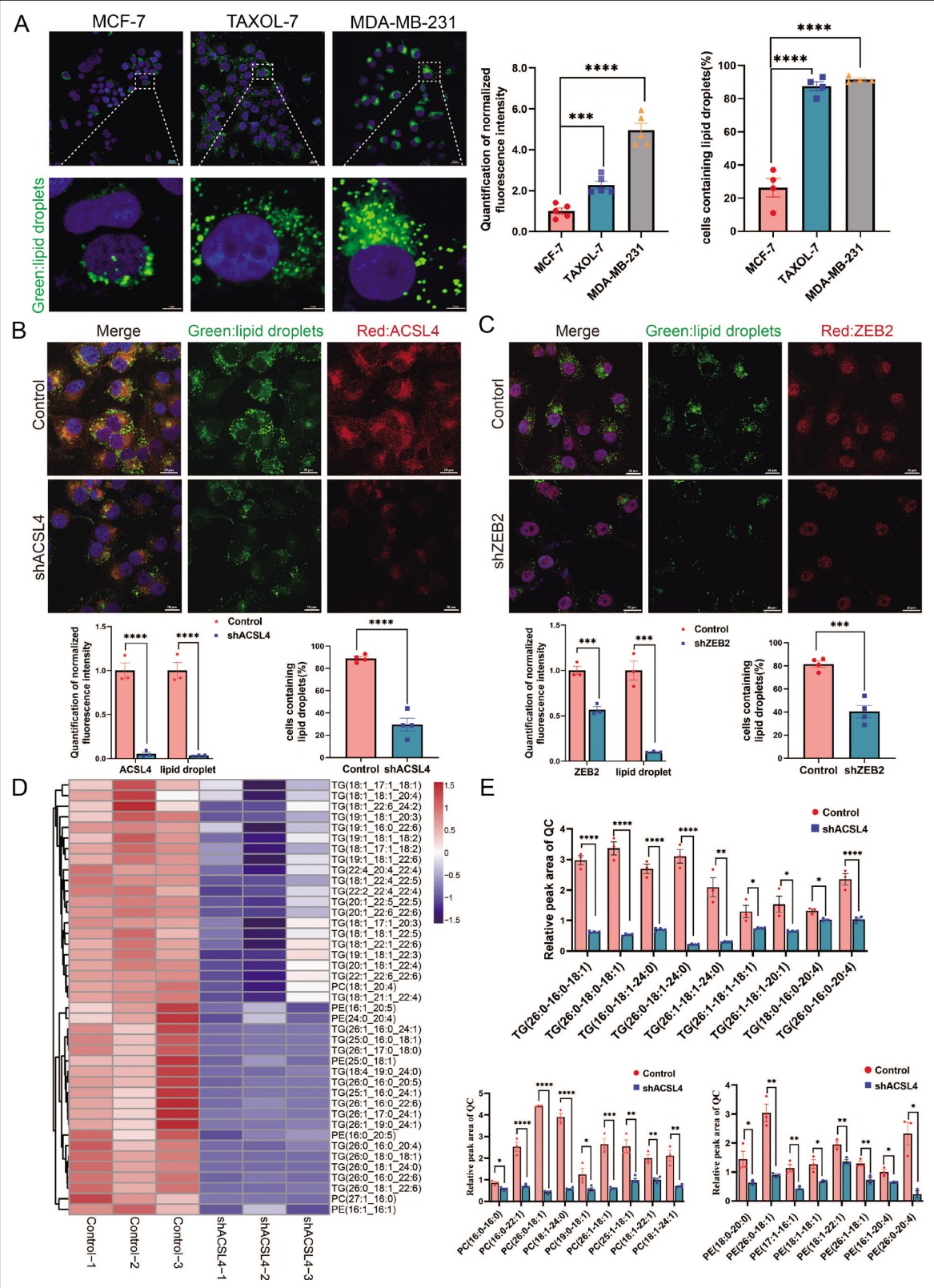

**Figure 3.** ACSL4 promotes lipid droplets (LDs) accumulation and lipogenesis. (**A**) BODIPY 493/503 staining of LDs in MCF-7, Taxol-resistant MCF-7 cells (TAXOL-7), or MDA-MB-231 cells (left panel). Quantification of normalized lipid contents (right panel). Scale bar, 5 µm. (**B**) BODIPY 493/503 staining of LDs in control or ACSL4 knockdown MDA-MB-231 cells (upper panel). Quantification of normalized fluorescence intensity and percentage of LDs-containing cell number (lower panel). Scale bar, 50 µm. (**C**) BODIPY 493/503 staining of LDs in control or ZEB2 knockdown MDA-MB-231 cells (upper

*Figure 3 continued on next page*

*Figure 3 continued*

panel). Quantification of normalized fluorescence intensity and percentage of LDs-containing cell number (lower panel). Scale bar, 50 µm. (**D**) The heatmap of representative downregulated lipid species (TG, PE, and PC) with hierarchical clustering in the control cells and ACSL4 knockdown cells. Each species was normalized to the corresponding mean value, as determined by two-way analysis of variance (ANOVA). (**E**) Quantification of different fatty acids containing TG, PC, and PE species in control or ACSL4 knockdown MDA-MD-231 cells. *$p < 0.05$, **$p < 0.01$, ***$p < 0.001$, ****$p < 0.0001$. Error bars, standard error of the mean (SEM).

The online version of this article includes the following source data and figure supplement(s) for figure 3:

**Source data 1.** The lipidomics data between control and ACSL4 knockdown MDA-MB-231 cells.

**Figure supplement 1.** Fluorescence staining of lipid droplets and ACSL4 in MDA-MB-231 cells.

**Figure supplement 2.** ACSL4 regulates lipid composition in basal-like breast cancer (BLBC) cells.

ZEB2-mediated metabolic process (*Figure 5H, I*). All these results indicated that ACSL4 upregulates CPT1A to stimulate FAO and ATP generation in BLBC cells.

## ZEB2 transcriptionally activates the expression of ACSL4

ZEB2 is a crucial transcription factor involved in EMT. Because ZEB2 and ACSL4 were highly correlated in the clinical samples in TCGA database, we investigated whether ACSL4 is a direct downstream target of ZEB2. Silencing of ZEB2 by siRNA (Small interfering RNA) in MDA-MB-231 cells significantly reduced ACSL4 mRNA levels and protein expression in MDA-MB-231 cells (*Figure 6A, B*). We found that the ACSL4 promoter contained four canonical ZEB2-binding E-boxes (CAGGT/CACCT) located at −287, −965, −1038, and −1116 of the ACSL4 promoter, respectively (*Figure 6C*). Therefore, we cloned five segments of the ACSL4 promoter and a control segment to generate promoter–luciferase constructs, based on the location of these E-boxes (*Figure 6C*). ZEB2 overexpression significantly enhanced the luciferase activity of all five E-box-containing segments of the ACSL4 promoter, whereas no significant change was observed in the control segment (*Figure 6D*). To examine whether ZEB2 directly binds to the ACSL4 promoter, we performed chromatin immunoprecipitation (ChIP) using four sets of primers (*Figure 6E*). Primer set 1, which covered segment 1, consistently exhibited apparent ZEB2 binding (*Figure 6F, G*). However, no binding was detectable using the other three sets of primers (*Figure 6F*), indicating that ZEB2 binds to the E-box located at nucleotides −184 to −295 of the ACSL4 promoter.

Since the −287 E-box exhibited apparent ZEB2 binding, we generated mutants of the −287 E-box promoter–luciferase construct. The luciferase reporter assay revealed that ACSL4 promoter activity was almost entirely abolished by the mutations (*Figure 6H*), suggesting that this promoter region is essential for ZEB2-mediated ACSL4 promoter activation. Taken together, these data suggest that ZEB2 directly binds to the ACSL4 promoter to activate its mRNA expression.

## ACSL4 regulates ZEB2 mRNA expression and protein stabilization

We hypothesized that ACSL4 regulates the expression of ZEB2. We then performed quantitative PCR and immunoblotting and observed that both ZEB2 mRNA and protein levels were reduced after the depletion of ACSL4 in the two BCSC cell lines (*Figure 7A, B*). A previous study has reported that ACSL4 regulates c-Myc protein stability in HCC. Our RNA-seq data revealed that some ubiquitin E3 ligases were significantly reduced in ACSL4 knockdown cells (*Figure 7—figure supplement 1*). We reasoned that ACSL4 might regulate the stability and ubiquitin of ZEB2. Therefore, we performed a ubiquitination assay to investigate whether ACSL4 regulates ZEB2 protein stability via ubiquitination. HEK293T cells were co-transfected with HA-ubiquitin and myc-ZEB2 expression vectors along with either an empty vector or an ACSL4 overexpression vector. As shown in *Figure 7C*, the expression of ACSL4 caused a significant decrease in the ubiquitination of ZEB2. Conversely, we observed a increasing ubiquitination of ZEB2 in ACSL4 silencing cells (*Figure 7D*), suggesting that ACSL4 attenuated the ubiquitin proteolysis of ZEB2. Notably, co-immunoprecipitation (Co-IP) assays and GST pull down assays revealed a specific interaction between ACSL4 and ZEB2 proteins, supporting the notion that ZEB2 and ACSL4 are present in a protein complex (*Figure 7E, F*, *Figure 7—figure supplement 2*). Immunofluorescence assay revealed that ACSL4 and ZEB2 were co-localized in some certain regions of the cytoplasm (*Figure 7—figure supplement 3*). To further confirm the role of ACSL4 in the regulation of ZEB2 proteolysis, cycloheximide was added to the cell medium to block the synthesis

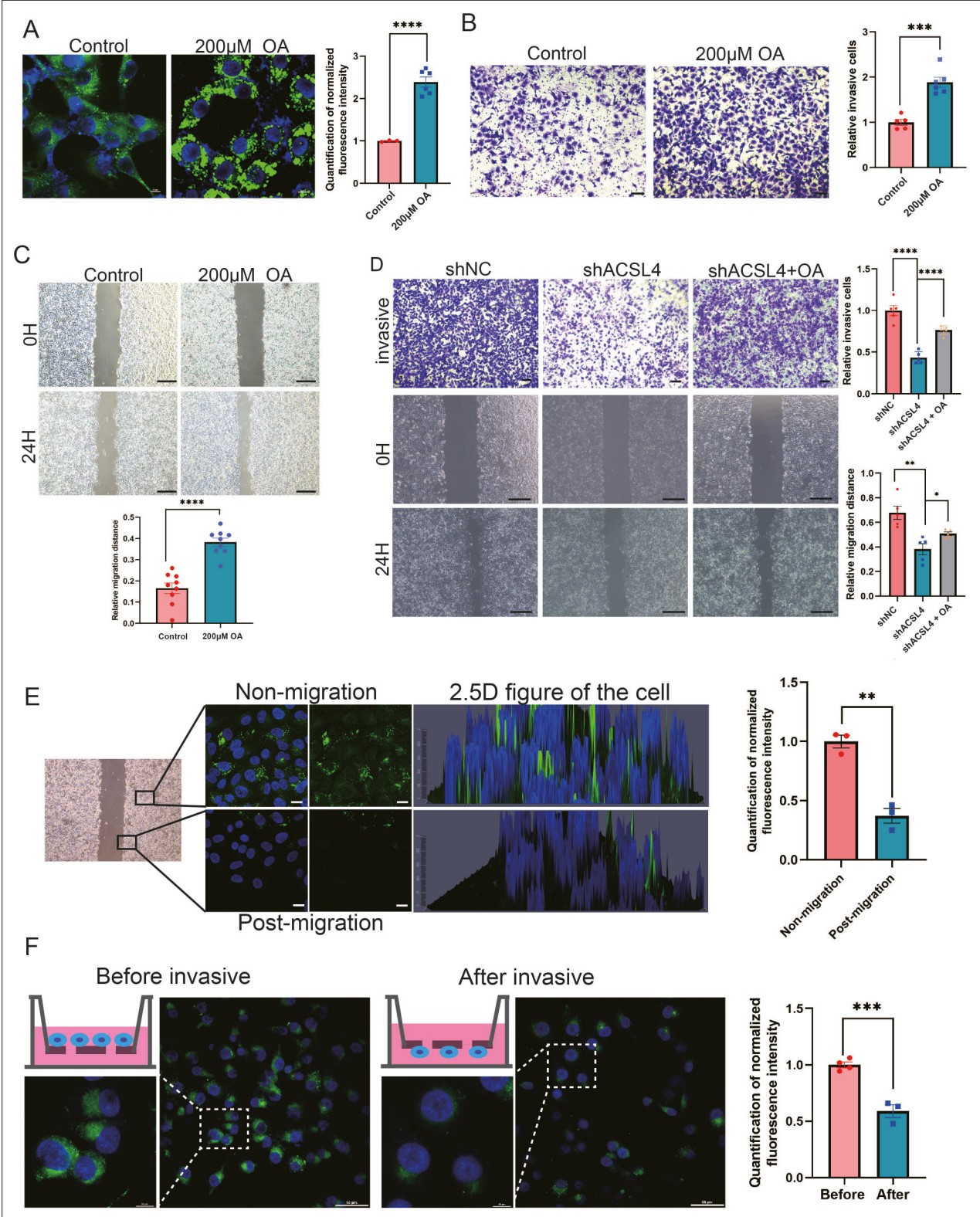

**Figure 4.** Exogenous lipids promote LDs accumulation and fuel cell migration. (**A**) BODIPY 493/503 staining of LDs in control cells or oleic acid (OA) loaded (200 µM) MDA-MB-231 cells. Scale bar, 10 µm. Quantification of normalized lipid contents from the conditions in (**A**). (**B**) Cell invasive capacity was analyzed by transwell invasion assay in control or OA loaded (200 µM) MDA-MB-231 cells. Cells invaded for 16 hr through a Matrigel-coated filter toward high-serum media (left panel). Quantification of relative invasive cells (right panel). Scale bar, 1 mm. (**C**) Cell metastatic capacity was analyzed by wound healing assays in control or OA loaded (200 µM) MDA-MB-231 cells. Scale bar, 5 mm. (**D**) Cell invasive and metastatic capacities were analyzed by

*Figure 4 continued on next page*

*Figure 4 continued*

transwell invasion assay and wound healing assays in control, shACSL4, and OA loaded ACSL4 knockdown MDA-MB-231 cells. Quantification of relative invasive and migrated cells. Scale bar, 1 /5 mm. (**E**) BODIPY 493/503 staining of LDs in the cells at the leading edge of the scratch and the cells that away from the edge. The 2.5 D figure of the cell was shown. Quantitation of total LDs area per cell. Scale bar, 10 μm. (**F**) Cells were seeded in a transwell chamber. After 24 hr, cells migrated to the lower side of the chamber, and the fluorescence intensity per cell was calculated (left panel). Quantitation of total LDs area per cell before and after cell migration (right panel). Scale bar, 10 μm (small) and 50 μm (big). *p < 0.05, **p < 0.01, ***p < 0.001, ****p < 0.0001.

The online version of this article includes the following figure supplement(s) for figure 4:

**Figure supplement 1.** Oleic acid (OA) induces cell migration in breast cancer cells.

of new proteins. Interestingly, endogenous ZEB2 protein levels were almost completely suppressed in ACSL4 knockdown cells from 0 to 8 hr time points (*Figure 7G, H*). This phenomenon could be explained by the fact that ZEB2 mRNA levels were significantly suppressed by the ACSL4 knockdown. Conversely, ZEB2 protein maintained a relatively steady level at 6 hr in ACSL4 overexpressing cells compared with control cells that undergone obvious ZEB2 proteolysis at 4 hr after adding CHX, indicating an increased ZEB2 protein stability in ACSL4 overexpressing cells (*Figure 7I, J*). Taken together, these results indicate that ACSL4 not only stabilizes the ZEB2 protein by attenuating its ubiquitination but also upregulates ZEB2 mRNA expression. Therefore, ACSL4 regulates ZEB2 through both transcriptional and post-transcriptional mechanisms.

## ACSL4 knockdown inhibits lung metastasis of BLBC in the animal model

To further validate our in vitro findings, we examined the effect of ACSL4 on lung metastasis of breast cancer cells in an animal model. Depletion of ACSL4 resulted in a significant reduction in tumor growth and lung colonization of highly metastatic MDA-MB-231 cells (*Figure 8A*). In contrast, the control group showed significantly more lung metastatic nodules (*Figure 8B*) and rapid tumor growth (*Figure 8C, D*). Immunohistochemistry confirmed a striking downregulation of ACSL4, accompanied by ZEB2 and vimentin, compared to those in the control group (*Figure 8E*).

To further validate the role of LDs in ACSL4 high-expression cells, we examined LDs content in frozen sections of the two groups. Consistent with the in vitro results, there was an obvious reduction in LDs accumulation in the ACSL4 depletion tissue compared to that in control tissue (*Figure 8F*). Taken together, these data suggest that ACSL4 promotes the lung metastasis of breast cancer cells in vivo.

## Discussion

The rewiring of metabolic pathways during EMT has only recently been recognized. In addition to glycolysis, dysregulated lipid metabolism has been shown to contribute to cancer invasion and metastasis (*Corbet et al., 2020*; *Soukupova et al., 2021*). Many EMT-driving factors have been found to reprogram the lipid metabolic pathways by regulating metabolic enzymes. For example, Transforming growth factor (TGF)-beta (TGF-β), a major driver of EMT, activates fatty acid synthase (FASN) and forms a FASN–TGFβ1–FASN-positive loop in cisplatin-resistant cells, resulting in EMT induction (*Jiang et al., 2015*; *Yang et al., 2016*; *Hua et al., 2020*). However, the direct regulation of lipid enzymes by EMT factors remains to be elucidated. Herein, we demonstrated that the lipid rating enzyme ACSL4 is a direct downstream target of the EMT transcription factor ZEB2 in controlling lipid metabolism. Mechanistically, ZEB2 activates ACSL4 mRNA expression by directly binding to its promoter, which contains four ZEB2 consensus sequences. Importantly, we observed a strong correlation between ZEB2 and ACSL4 expression levels in clinical breast cancer samples. ACSL4 re-expression rescues the migration ability of ZEB2-depleted cells. These observations indicated that ACSL4 is crucial for ZEB2-mediated metastasis.

We also demonstrated that ACSL4 directly binds to and stabilizes ZEB2 by reducing its ubiquitination. Our results are consistent with a recent report that ACSL4 stabilizes the oncoprotein c-Myc via the ubiquitin–proteasome system (*Chen et al., 2020*). We propose that as an LDs enzyme, ACSL4 may participate in the protein degradation system of LDs. This is likely to be a novel function of ACSL4. Interestingly, a recent study reported that LDs recruit Numb through the AP2A/ACSL3 complex to promote Numb degradation (*Liu et al., 2022*). Thus, it is likely that LDs act as a platform for protein

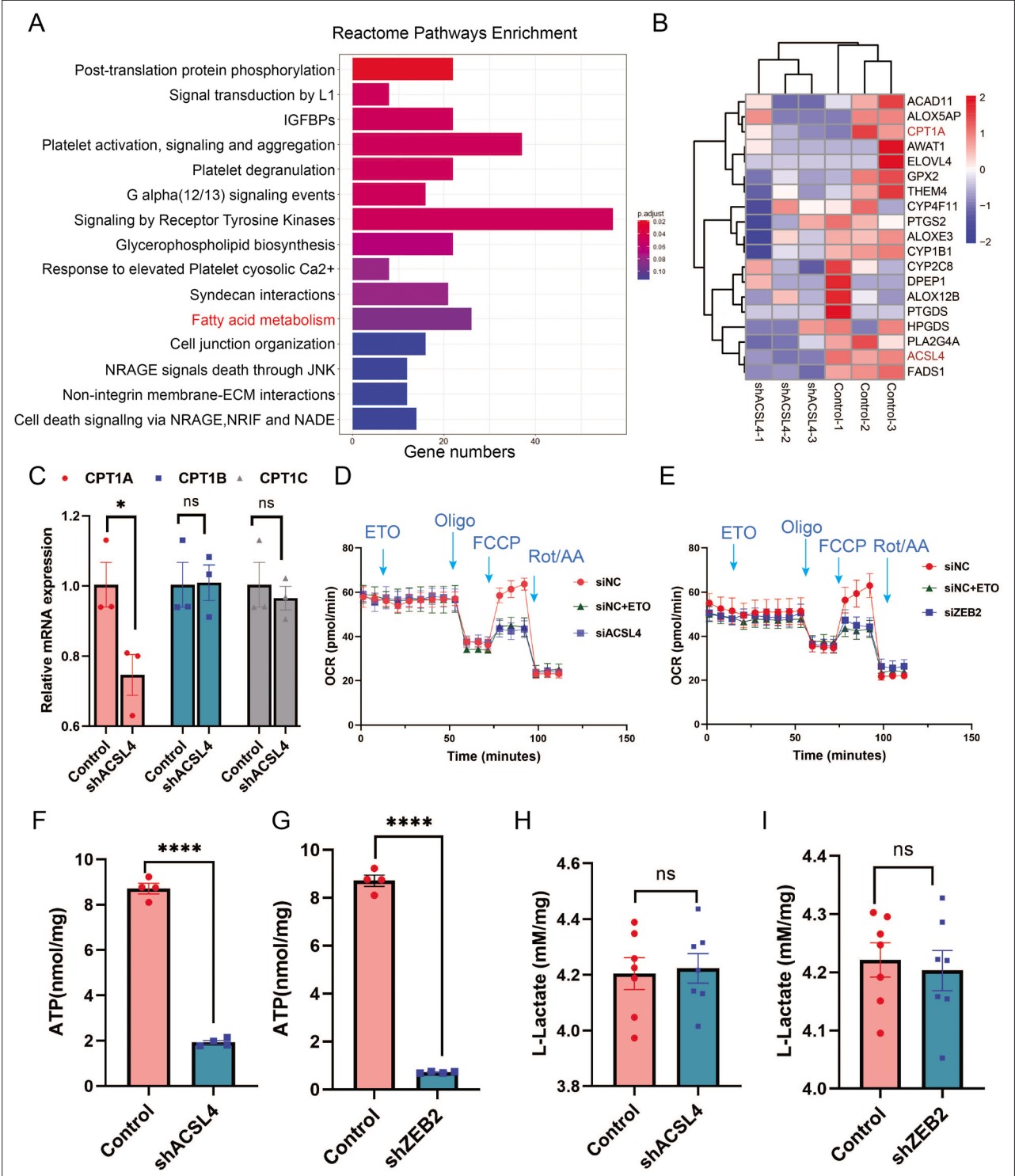

**Figure 5.** ACSL4 and ZEB2 increase fatty acid oxygen consumption and promote adenosine triphosphate (ATP) generation in basal-like breast cancer (BLBC) cells. (**A**) Reactome pathway analysis of differentially expressed genes by RNA-sequencing between control and ACSL4 knockdown MDA-MB-231 cells. The representative pathways were shown. (**B**) Heatmap showing the differentially expressed genes between control cells and ACSL4 knockdown cells. The representative fatty acid oxidation (FAO)-related genes were shown. (**C**) The mRNA levels of CPT1A, CPT1B, and CPT1C were analyzed by quantitative PCR in control and ACSL4 knockdown cells. (**D**) Quantitation of the normalized oxygen consumption rate (OCR) for long-chain fatty acids was monitored by Agilent XF Substrate Oxidation Stress Test in control or ACSL4 knockdown cells. Specific inhibitors were added as indicated. (**E**) Quantitation of the normalized OCR for long-chain fatty acids was monitored by Agilent XF Substrate Oxidation Stress Test in control or ZEB2 knockdown cells. Specific inhibitors were added as indicated. (**F, G**) ATP production was quantified in control or ACSL4 knockdown cells (left) or

*Figure 5 continued on next page*

*Figure 5 continued*

ZEB2 knockdown cells (right). (**H, I**) Lactate production was examined in control and ACSL4 knockdown (shACSL4) or ZEB2 knockdown (shZEB2) cells. Data are represented as mean ± standard error of the mean (SEM) of three independent experiments, analyzed by Student's *t*-test, *p < 0.05, ****p < 0.0001.

The online version of this article includes the following source data and figure supplement(s) for figure 5:

**Source data 1.** The raw unedited gels or blots images of *Figure 5*.

**Figure supplement 1.** ACSL4 regulates the expression of lipid metabolic genes.

degradation. Interestingly, our RNA-seq data revealed that some ubiquitin E3 ligases, such as FBXO4, UBE3C, NEDD4, and RBX1 were significantly reduced in ACSL4 knockdown cells (*Figure 7—figure supplement 1*). This result indicated that ACSL4 may reduce the ubiquitin of ZEB2 via downregulating ubiquitin E3 ligase. Additionally, we found that ACSL4 promoted ZEB transcription as the mRNA level of ZEB2 was significantly reduced after ACSL4 knockdown. A recent study reported that LDs-derived lipolysis provide acetyl-CoA for the epigenetic regulation of gene transcription (*Ippolito et al., 2022*). We observed that ACSL4 can also promote FAO, which generates acetyl-CoA for the epigenetic regulation. It is likely that ACSL4 regulates the ZEB2 mRNA level via lipid-epigenetic reprogramming mechanism, which is worth studying in the future. Therefore, ACSL4 regulates ZEB2 not only via a post-transcriptional mechanism but also via a transcriptional mechanism. Notably, the relationship between ZEB2 and ACSL4 could form a positive feedback loop: ZEB2 transcriptionally activates ACSL4, and conversely, ACSL4 stabilizes the ZEB2 protein by reducing its ubiquitination. Amplification of either gene might therefore lock this loop in an active state, resulting in the enhanced invasive and metastatic capabilities of breast cancer cells (*Figure 8G*).

Previous studies have reported that ACSL4 could act as a tumor suppressor or oncogene, depending on the specific cancer type and tissue environment (*Cheng et al., 2020*; *Ma et al., 2021*; *Lei et al., 2020*; *Zhang et al., 2021*; *Grube et al., 2022*). Indeed, ACSL4 could either be located at the cytomembrane or at the LDs and endoplasmic reticulum membrane, indicating the different functions of ACSL4 (*Quan et al., 2021*). The cytomembrane ACSL4 is likely responsible for dictating ferroptosis sensitivity by shaping the plasma membrane lipidome, whereas ACSL4 localized to LDs has other functions. We provide evidence that ACSL4, which is located in LDs, is a pro-metastatic factor that promotes invasion and migration. Notably, we demonstrated that ACSL4 depletion significantly suppressed the invasion and migration of breast cancer cells in vitro and in vivo. Furthermore, ACSL4 and ZEB2 were found to be preferentially expressed in BLBC cells and clinical samples that lacked ERα expression. Survival analysis revealed that breast cancer patients with high expression of both ACSL4 and ZEB2 are associated with worse overall survival than those patients with low expression. Thus, ACSL4 and ZEB2 could specifically be used as prognostic markers for breast cancer, and this axes hold promise as a new metabolic therapeutic target for highly invasive breast cancer.

Although ACSL4 has been shown to play an essential role in metastasis in many types of cancer (*Wu et al., 2015*; *Maloberti et al., 2010*; *Sánchez-Martínez et al., 2017*), its mechanism of ACSL4-mediated metastasis is not fully understood. ACSL4 is a member of the long-chain fatty acetyl-CoA synthetase enzyme family that catalyzes FAs to their active form, acyl-CoA, which can be directed to anabolism or catabolism, depending on the cellular background. We observed that the knockdown of ACSL4 significantly reduced the number and size of LDs, indicating that FAs were directed to anabolism to form LDs by ACSL4. Indeed, it has been shown that fatty acids are stored in LDs before entering FAO, and the formation of LDs is required for FAO (*Li et al., 2020*; *Petan, 2020*; *Welte and Gould, 2017*). This is in line with our observation that exogenous FAs (OA) significantly enhance LDs accumulation and migration. Importantly, lipidomic analysis revealed that the knockdown of ACSL4 significantly decreased the incorporation of both saturated and unsaturated FAs into different species of lipids, including TAG, phospholipids (PE and PC), and cholesteryl esters (CE). Consistently, frozen tissue sections from a metastatic animal model confirmed that high ACSL4 expression was accompanied by apparent LDs accumulation. These results suggest that ACSL4 is a crucial regulator that promotes lipid storage during breast cancer metastasis. Interestingly, knockdown of ACSL4 restored the expression of E-cadherin, an essential adhesion molecule, indicating that ACSL4 promotes tumor invasion through multiple mechanisms.

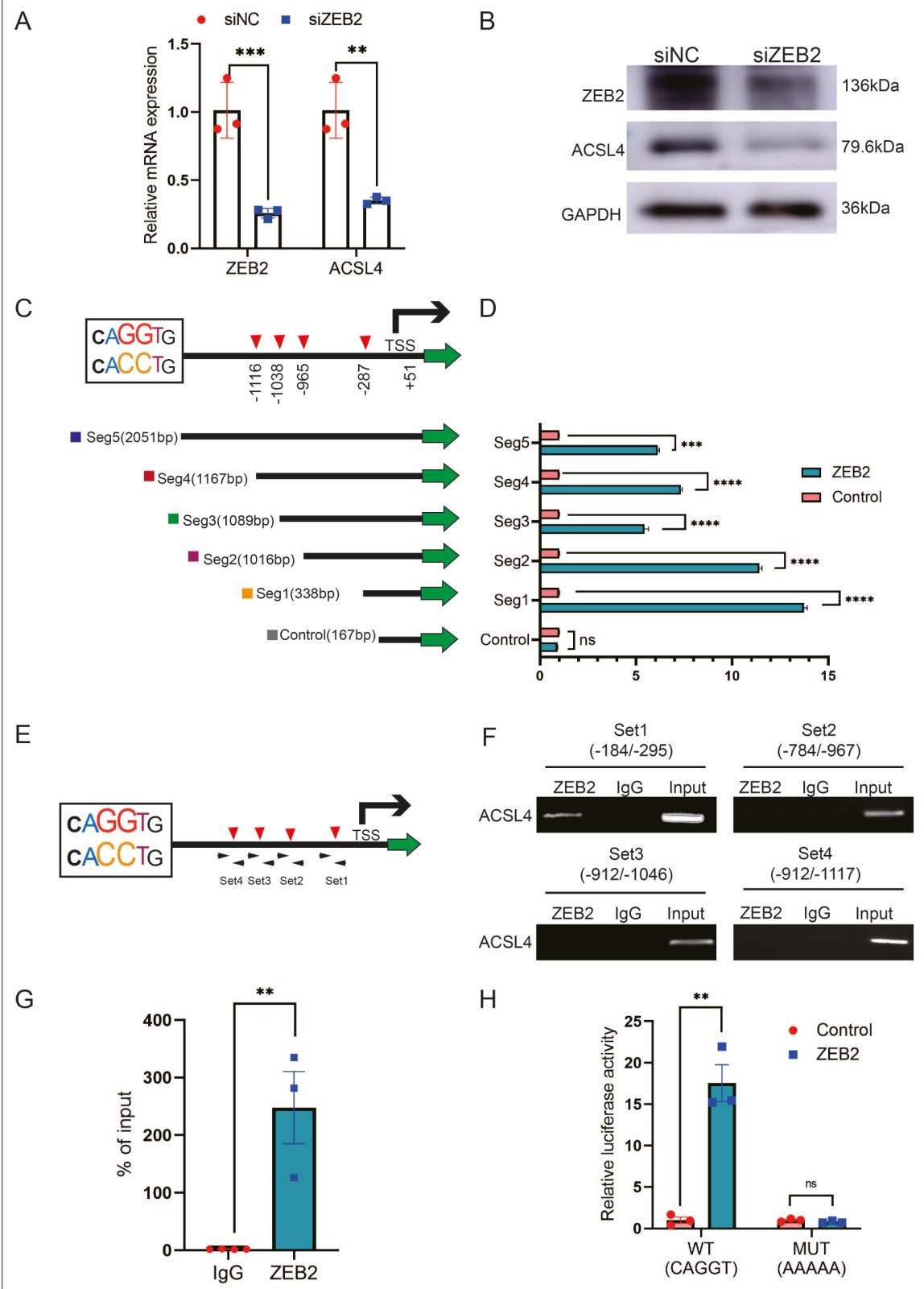

**Figure 6.** ZEB2 activates ACSL4 expression by directly binding to its promoter. (**A**) Relative mRNA levels of ZEB2 and ACSL4 in control or ZEB2 knockdown MDA-MB-231 cells. (**B**) Protein levels of ZEB2 and ACSL4 in control or ZEB2 knockdown MDA-MB-231 cells. (**C, D**) Truncated ACSL4 promoter segment activity analyzed by luciferase reporter assay in control or ZEB2 overexpressed 293T cells. (**E**) The specific primers designed for ACSL4 promoter according to the E-box position shown in C (set1 to −287 bp, set2 to −965, set3 to −1038, set4 to −1116). (**F**) Chromatin

*Figure 6 continued on next page*

*Figure 6 continued*

immunoprecipitation (ChIP) assay analysis of the occupation of ZEB2 on ACSL4 promoter by using four primers as indicated in E. (**G**) Quantitative PCR analysis of ZEB2-binding abundance of specific ACSL4 promoter region by using set1 primer indicated in F. Genomic DNA was purified after ChIP and analyzed by quantitative PCR. (**H**) Luciferase reporter analysis of the activity of wild-type ACSL4 promoter or its mutants in control or ZEB2 overexpression 293T cells. Data are represented as mean ± standard error of the mean (SEM) of three independent experiments, analyzed by Student's *t*-test, \*\*p < 0.01, \*\*\*p <0 .001, \*\*\*\*p < 0.0001, ns: no significance.

The online version of this article includes the following source data for figure 6:

**Source data 1.** The raw unedited gels or blots images of *Figure 6*.

Reprogramming of lipid metabolism is an essential step in metastasis. LDs are highly dynamic monolayer membrane-bound organelles involved in energy utilization, signal transduction, and cancer invasion. The number and size of LDs are associated with cancer aggressiveness (*Cruz et al., 2020*). We observed that highly invasive breast cancer cells were enriched with LDs. Using fluorescence microscopy, we noticed that the number of LDs was significantly reduced after cell migration. Our findings are in line with a recent study showing that LDs undergo lipolysis during the process of migration in pancreatic cancer (*Rozeveld et al., 2020*). It is likely that FAs are released from LDs and form acyl-CoA, which enters OXPHOS to support the energy needed for metastasis. Therefore, these data support the notion that increased LDs accumulation is a priming state prior to metastasis, and that LDs are crucial resources to fuel the process of metastasis.

In addition, we observed that ACSL4 also participates in FA catabolism, as the long-chain FA-derived OCR and ATP production were significantly reduced in ACSL4-depleted cells, indicating that ACSL4 is essential for FAO stimulation. Our results are in line with a recent study that reported that hexokinase 2 enhances tumorigenicity by activating the ACSL4-mediated FA β-oxidation pathway (*Li et al., 2022*). Notably, the expression of CPT1, a rate-limiting enzyme of FAO, was significantly reduced in the ACSL4-depleted cells. Among the three isoforms of CPT1, CPT1A is the only isoform regulated by ACSL4. We proposed that ACSL4 promotes FAO and ATP production by upregulating CPT1A, thereby providing energy support for breast cancer metastasis. Our results reveal the mechanism of previous findings that FAO is a critical energy pathway in TNBC (*Camarda et al., 2016*). Furthermore, we observed that lactate levels did not change after ACSL4 knockdown, suggesting that glycolysis is not involved in ACSL4-mediated energy metabolism. Therefore, we propose a model in which cell migration is a multistep process accompanied by dynamic cellular lipid metabolic changes. At the pre-metastatic stage, ACSL4 enhances LDs accumulation by promoting lipogenesis. During metastasis, ACSL4 stimulates FAO to generate ATP.

In conclusion, we provide insights into the mechanistic links between EMT and lipid metabolism and identify the ZEB2/ACSL4 axis as a novel metastatic metabolic pathway that stimulates both lipogenesis and FAO, resulting in enhanced breast cancer invasion and metastasis. Importantly, our results demonstrate that ACSL4 is a direct downstream target of ZEB2 in controlling lipid storage and LDs accumulation, which are important steps and energy pools for metastasis. Clinically, our findings identified ZEB2–ACSL4 signaling as an attractive therapeutic target for overcoming breast cancer metastasis. Elevated ACSL4 levels can be used as an effective marker for predicting cancer progression in patients with advanced breast cancer. The limitation of this study is the clinical samples is only 45. The future study should expand the clinical samples and cases to provide more clinical evidence for the crucial role of ACSL4 in breast cancer metastasis.

## Materials and methods
### Breast cancer cell lines and clinical specimens

Breast cancer cell lines MCF-7, MDA-MB-231, and BT549 were purchased from the American Type Culture Collection (ATCC, Beijing Zhongyuan Ltd, Beijing, China). Paclitaxel- and epirubicin-resistant cell lines have been reported previously (*Yang et al., 2017*; *Zheng et al., 2014*). All cell lines were maintained in Dulbecco's modified Eagle's medium (Gibco) supplemented with 10% (vol/vol) fetal bovine serum (FBS; BI, Biological Industries) and 1% (vol/vol) Pen/Strep (Gibco) and incubated in a humidified atmosphere of 5% $CO_2$ at 37°C. Cell were regularly tested for mycoplasma contamination using commercially available Mycoplasma Detector kit (MycoBlue kit, Vazyme). Cell lines were authenticated using STR profiling (IGEbio, Inc).

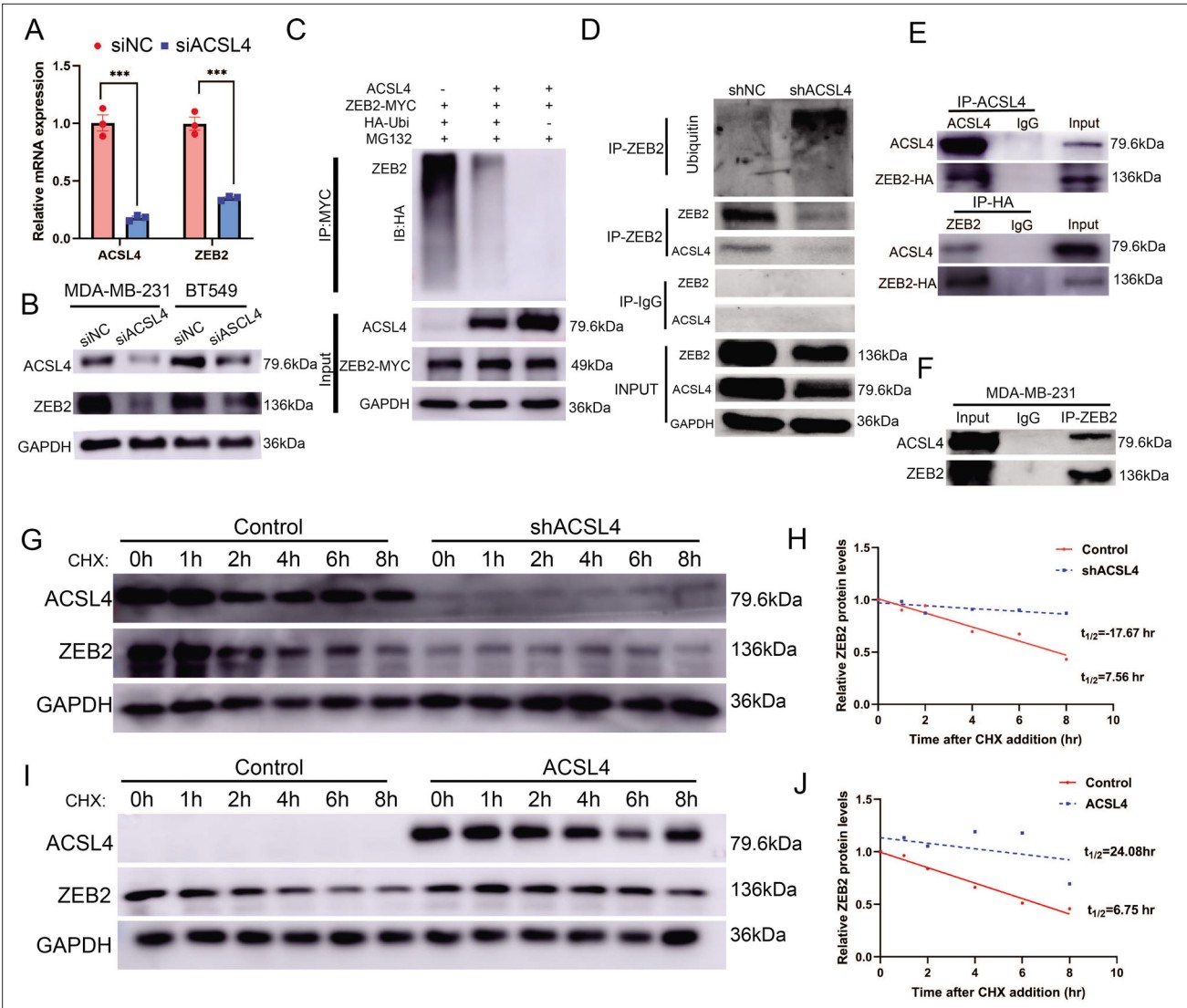

**Figure 7.** ACSL4 regulates ZEB2 mRNA expression and protein stability. (**A**) Relative mRNA levels of ZEB2 and ACSL4 in control or ACSL4 silencing MDA-MB-231 cells. (**B**) Protein levels of ZEB2 and ACSL4 in control or ACSL4 silencing in MDA-MB-231 cells and BT549 cells as indicated. (**C**) Ubiquitylation of ZEB2 was examined by in vitro ubiquitin assay. 293T cells were co-transfected with indicated constructs. Cells were treated with MG132 for 6 hr before IP. Anti-MYC was used to pull down the ZEB2 protein. The polyubiquitinated ZEB2 protein was detected by an anti-HA antibody. (**D**) Ubiquitylation of ZEB2 was examined in MDA-MB-231 cells. The indicated antibody was used to pull down the protein in control or ACSL4 knockdown MDA-MB-231 cells. The polyubiquitinated ZEB2 protein was detected by anti-ubiquitin antibody. (**E**) The interaction between ACSL4 and ZEB2 was detected by co-immunoprecipation (Co-IP) assay. 293T cells were co-transfected with ZEB2 and ACSL4 expressing construct. The indicated antibody was used to pull down the protein. (**F**) The interaction between ACSL4 and ZEB2 was detected by IP assay in MDA-MB-231 cells. Anti-ZEB2 antibody was used to pull down the protein. (**G**) The stability of ZEB2 protein was detected by CHX treatment assay in control or ACSL4 silencing MDA-MB-231 cells. Cells were treated with 100 μg/ml cycloheximide (CHX) and were harvested at the indicated times after the addition of CHX. GAPDH was used as the internal loading control. (**H**) Quantification of stability assays shown in G. (**I**) The stability of ZEB2 protein was detected by CHX treatment assay in control or ACSL4 overexpressed MCF-7 cells. GAPDH was used as the internal loading control. (**J**) Quantification of stability assays shown in I. ***p < 0.001.

The online version of this article includes the following source data and figure supplement(s) for figure 7:

**Source data 1.** The raw unedited gels or blots images of *Figure 7*.

**Figure supplement 1.** The differentially expressed genes between control cells and ACSL4 knockdown cells were shown.

**Figure supplement 2.** GST pull down assay was used to examine the interaction between ACSL4 and ZEB2 in MDA-MB-231 cells.

**Figure supplement 3.** Fluorescence staining of ACSL4 and ZEB2 in MDA-MB-231 cells.

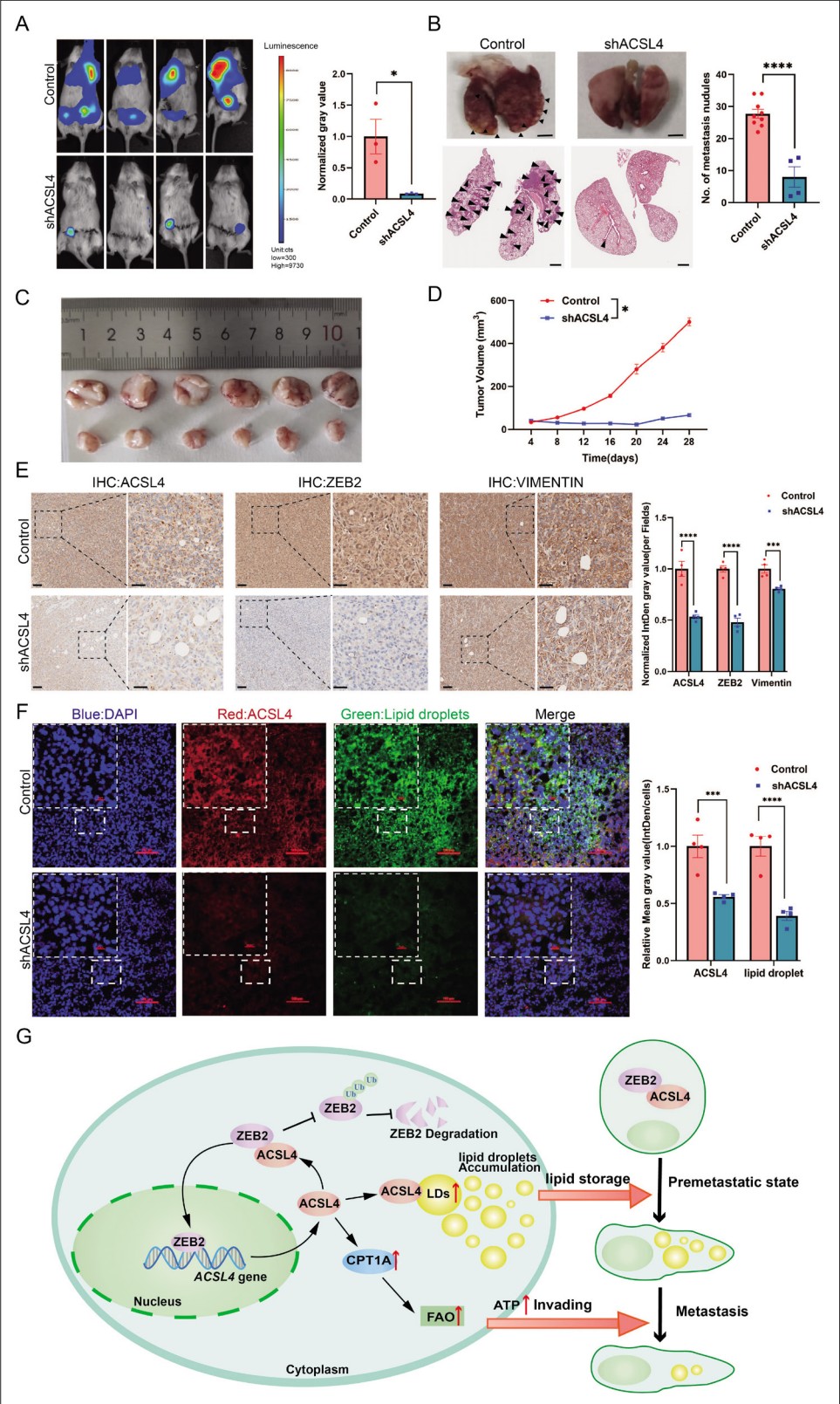

**Figure 8.** ACSL4 knockdown attenuates lung metastasis of breast cancer. (**A**) Cell metastatic capacity was determined by xenograft experiment in vivo. Control cells and ACSL4 knockdown MDA-MB-231 cells were mixed with Matrigel in a 1:1 ratio and injected into NSG mice. Lung metastatic burden in mice was quantified by bioluminescence imaging at the experimental endpoint (left panel). Quantification of normalized gray area (right

*Figure 8 continued on next page*

*Figure 8 continued*

panel). (**B**) Representative xenograft tumor images and HE staining pictures of metastatic nodules in lungs (pointed by black arrows) were shown. The right panel shows the quantification of metastatic lung nodules in two groups as indicated. (**C**) The image of xenograft tumors developed from the control and ACSL4 knockdown cells. (**D**) The tumor volumes were measured and calculated at the indicated time. (**E**) Representative immunohistochemistry (IHC) images of lung metastatic lesions (left panel) and quantification is shown (right panel). Scale bar, 50 µm. (**F**) The representative images of fluorescence assay for ACSL4 and lipid droplets (left panel). The fluorescence intensity of lipid droplets and ACSL4 were calculated in the right panel. Scale bar, 100 µm. (**G**) A proposed mechanism to illustrate the positive feedback loop of ZEB2 and ACSL4 regulates lipid metabolism, which results in enhanced breast cancer metastasis. Data are represented as mean ± standard error of the mean (SEM) of three independent experiments and analyzed by Student's *t*-test, *p < 0.05, ***p < 0.001, ****p < 0.0001.

Primary breast cancer and adjacent normal tissues were collected from the volunteers at The Second Affiliated Hospital of South China University of Technology (Guangzhou First People's Hospital), Guangdong, China. This study was approved by the Ethics Committee of The Second Affiliated Hospital of South China University of Technology (Guangzhou First People's Hospital) (approval no. K2021-201-01). Forty-five female patients were enrolled in this study. The specimens were then subjected to western blotting and immunohistochemistry (IHC) assays.

## Plasmid construction and cell transfection

The PLKO.1-TRC-LUC lentivirus vector containing shRNA (shACSL4-1 and shACSL4-2) and the control vector were purchased from WZ Biosciences. The virus package plasmids psPAX2 and pMD2.G were purchased from Addgene. To produce the lentivirus, the shRNA vector and package plasmids were co-transfected into 293T cells using polyethyleneimine (Polysciene). After 48 or 72 hr, supernatants were harvested and passed through 0.45 µm filters to collect the virus and stored at −80°C until further use. MDA-MB-231 cells were transfected with a viral solution containing 8 µg/ml polybrene (Hanbio Biotechnology). Forty-eight hours after transfection, the virus was removed from the culture and fresh medium was added. The cells were subsequently selected using 2 mg/ml puromycin to obtain a stable cell line. MCF-7 cells were transfected with HBLV-h-ACSL4-3xflag-ZsGreen-PURO or control lentiviral plasmid. The plasmids were purchased from Hanbio Biotechnology.

For the siRNA assay, 5 nM negative control siRNA or ACSL4 and ZEB2 siRNA (GenePharma) were transfected into MDA-MB-231 or paclitaxel-resistant cells using Lipofectamine 3000 (Invitrogen) according to the manufacturer's instructions. Six hours after transfection, the medium was replaced with fresh growth medium and the cells were harvested after 24–48 hr for qPCR (Quantitative PCR)or western blot analysis. The siRNA sequences are listed in *Supplementary file 1*.

## Western blot analysis

For western blotting, cells were lysed in WIP buffer (Beyotime) containing protease and phosphatase inhibitors. Protein levels were quantified by BCA assay (Beyotime), and equal amounts were separated using 10% sodium dodecyl sulfate–polyacrylamide gel electrophoresis (SDS–PAGE) and transferred onto a 0.22-µm Polyvinylidene fluoride (PVDF) membrane for probing. After blocking for 1 hr in 5% non-fat milk diluted in TBST, the membrane was incubated with various primary antibodies at 4°C overnight. The following primary antibodies were used: ACSL4 (ab155282, Abcam), ZEB2 (sc-271984, Santa Cruz), E-cadherin (3195S, CST), N-cadherin (610921, BD Biosciences), vimentin (D21H3, 5741T, CST), CPT1A (ab220789, Abcam), GAPDH (10494-1-AP, Protein Tech), Myc-Tag (71D10, 2278T, CST), and HA-Tag (C29F4, 3724T, CST). Next, secondary antibodies were added and the proteins were detected using an ECL kit (Millipore). Immunoreactive signals were visualized using the GE Amersham Imager 600 chemiluminescence system.

## Migration and invasion assay

A wound healing assay was performed to examine the migratory ability of the cells. Briefly, cells were seeded in 6-well plates at a density of $4 \times 10^5$ cells per well in a complete medium at 37°C. Until cells reached 80–90% density, a sterile plastic tip was used to create a wound line across the surface of the plates. The suspended cells were discarded. After the cells were cultured in reduced serum

Dulbecco's modified Eagle medium in a 5% $CO_2$ incubator at 37°C for 48 hr. Images were obtained using a phase-contrast microscope. Each assay was performed in triplicates.

The cell invasion capacity was measured using Matrigel-coated transwell chambers (8.0 μm; Corning Inc). Briefly, cells were suspended in 200 μl serum-free medium and seeded in the upper chamber of a transwell. The lower chamber of the transwell was filled with medium containing 10% FBS-containing medium. The transwell chambers were then incubated at 37°C. After culturing for 24 hr, the transwell holes were penetrated with 4% paraformaldehyde for 20 min and then stained with 0.1% crystal violet solution for 15 min. Invading cells were imaged and counted in five random fields.

### Immunofluorescence and IHC assays

For the immunofluorescence assay, cells were seeded on confocal dishes, washed in phosphate-buffered saline (PBS) three times, and fixed in 4% paraformaldehyde for 15 min at room temperature. The cells were blocked with 10% goat serum (AR0009, BOSTER) for 1 hr at room temperature and washed three times. The cells were then incubated with the primary antibodies for 1 hr at room temperature. Subsequently, the cells were rinsed thrice with PBS and incubated with secondary antibodies for 1 hr at room temperature.

For BODIPY 493/503 and Phalloidin staining, cells were fixed in 4% paraformaldehyde for 15 min and washed twice with PBS. The cells were then incubated with BODIPY 493/503 (1:2500 in PBS, GLPBIO)/Phalloidin (HUAYUN) for 15 min at room temperature. DAPI (4',6-diamidino-2-phenylindole) was used to stain the nuclei. Images were acquired by confocal microscopy (Ni-E-A1, Nikon, ×40) and analyzed using NIS Elements Viewer software and ImageJ.

For IHC assay, paraffin-embedded tissue slides were dewaxed with xylene and rehydrated using a graded series of alcohols. This was followed by antigen retrieval and blocking with 5% bovine serum albumin for 60 min. After that, tissue slices were incubated with primary antibodies against ACSL4, ZEB2, vimentin, or ERα at 4°C overnight in a humidified container and then detected with the SP Rabbit&Mouse HRP Kit (CWBIO). Images were acquired using a digital pathology scanning system (Aperio CS2).

### Metabolic analysis

For the OCR assay, the XF long-chain fatty acid oxidation (LCFA) stress test kit (103672-100) and Seahorse XF96 Analyzer were used to investigate the long-chain FA OCR and extracellular acidification rate according to the manufacturer's protocol.

ATP levels were determined using an ATP Assay Kit (S0026, Beyotime) according to the manufacturer's protocol. ATP levels were calculated using luminescence signals and were normalized to protein concentrations. Glycolytic activity was determined using a Glycolysis Cell-Based Assay Kit (600450, Cayman Chemical), according to the manufacturer's protocol.

### Untargeted lipidomics and proteomics

Cell lipids were extracted in a chloroform–methanol mixed solution (2:1, −20°C). The extracted cell lysates were immersed in liquid nitrogen and frozen for 5 min. The extracted cell lysates were placed in a 2 ml adapter, and the above steps were repeated twice. Samples were then centrifuged to pellet the proteins (5 min, 12,000 rpm), and the supernatant was stored for analysis in a vacuum centrifugal concentrator. The sample was dissolved in 200 μl isopropanol, filtered through a 0.22-μm membrane, and detected by liquid chromatography–mass spectrometry. The lipidomics data were analyzed using LipidSearch software (version 4.0). The software identified intact lipid molecules based on their molecular weight and fragmentation pattern using an internal library of predicted fragment ions per lipid class. The spectra were then aligned based on retention time and the MS1 peak areas were quantified across the sample conditions. Excel 2010 was used to calculate intensity, and the R program (version 3.2.5) was used for data manipulation and statistical analyses, including unsupervised hierarchical clustering and heatmap visualization.

### Quantitative real-time PCR

Total RNA was isolated using the TRIzol reagent (Invitrogen), and cDNAs was synthesized from total RNA using the PrimeScript RT Master Mix (Perfect Real Time, Takara). Quantitative real-time PCR

was performed in triplicate using PowerUp SYBR Green (Thermo Fisher Scientific). The relative gene expression was measured using the $2^{-\Delta\Delta Ct}$ method. All primers used are listed in *Supplementary file 2*.

## Luciferase reporter assays

For luciferase assays, the 2000 bp ACSL4 promoter vector containing luciferase was purchased from WZ Biosciences Inc. The −287, −965, −1038, and −1116 bp regions of the ACSL4 promoter were cloned by PCR amplification using primers containing restriction sites MIII and Hind III. The nucleotide sequences of primers used are listed in *Supplementary file 3*. 293T cells were transfected with 1 μg of the −287, −965, −1038, −1116, or 2000 bp ACSL4 promoter luciferase reporter and an empty vector for 24 hr. After transfection, the cells were harvested and analyzed using Bright-Glo reagent (Promega) according to the manufacturer's instructions.

## ChIP assay

ChIP assays were performed using a ChIP assay kit (Cat.p2078, Beyotime Biotechnology). Briefly, MDA-MB-231 cells were grown to 90% confluence and crosslinking was performed with 1% formaldehyde for 10 min. Mouse anti-ZEB2 antibody or mouse IgG was used to immunoprecipitate the DNA-containing complexes. After the DNA purification (Cat. D0033; Beyotime Biotechnology), PCR, and qPCR were performed to detect the ZEB2-binding site in the ACSL4 promoter region. The primer sequences are listed in *Supplementary file 4*.

## Co-IP assay and ubiquitination assay

For Co-IP assays, HEK293T cells were co-transfected with ACSL4-FLAG and ZEB2-MYC plasmids. Forty-eight hours after transfection, cells were lysed in WIP buffer (Beyotime) containing a protease inhibitor and phosphatase inhibitors for 30 min at 4°C followed by centrifugation. The supernatants were immunoprecipitated with the indicated antibodies overnight at 4°C, followed by incubation with protein A/G beads for 1 hr at 4°C. After incubation, the beads were washed with WIP buffer and boiled in a 2× loading buffer. Protein samples were analyzed by western blotting. GST pulldown assay was performed by using GST pull down Assay Kit (FI88807, FITGENE) according to the manufacturer's protocol. The GST control plasmid and ACSL4-GST plasmid were purchased from WZ Biosciences.

For the ubiquitination assay, HEK293T cells were transfected with HA-Ubi plasmid, ACSL4-FLAG plasmid, and ZEB2-MYC or vector plasmid. Forty-eight hours after transfection, the cells were treated with 20 μM MG-132 for 6 hr to block the proteasomal degradation of ZEB2 before being lysed with WIP lysis buffer (Beyotime). Equal amounts of protein lysates were immunoprecipitated with anti-MYC beads and subjected to SDS–PAGE, followed by blotting with anti-HA (ubiquitin) to visualize polyubiquitinated ZEB2 protein bends.

## In vivo experiments

The Ethics Committee approved the animal experiments for Animal Experiments of the South China University of Technology (The Animal Ethics Committee Number is 2020054). All NSG mice were purchased from the Medical Laboratory Animal Center of Guangdong Province, China. Six female NSG mice aged 6–8 weeks were used in each group for the primary tumor growth and spontaneous lung metastasis experiments. A total of $2 \times 10^6$ vector control or shACSL4 cells were mixed 1:1 by volume with Matrigel (BD Biosciences). Each mouse was injected orthotopically into both the flanks. Xenograft tumor growth was measured and tumor volume was calculated as follows: volume = (length × width$^2$)/2. At the experimental endpoint, mice were intraperitoneally injected with 150 mg/kg D-luciferin and imaged for 2 min using a live imager. The xenograft tumor and lung samples were removed and fixed to count the metastatic lung nodules.

## Statistical analysis

The results are reported as the mean ± standard error of the mean, as indicated in the figure legend. Student's *t*-test was used for two-group comparisons. Comparisons between three and more groups were analyzed by one-way analysis of variance followed by Duncan's test. Statistical comparisons for the LM2 lung metastasis assay were performed using the Mann–Whitney *U*-test. All experiments with representative images, including western blotting and immunofluorescence, were repeated at least twice, and representative images are shown. p < 0.05 was considered statistically significant.

## Acknowledgements

This research was supported by the Guangzhou Science and Technology Project (202201020240 to N Yang), the Medical Research Foundation of Guangdong Province (A2021112 to N Yang), the National Natural Science Foundation of China (81602434 to N Yang), the Research Agreement between South China University of Technology and Guangzhou First People's Hospital (D9194290 to Y Duan), the National Natural Science Foundation of China (81972594 to M Yan), and the Guangzhou Science and Technology Project (2023A04J0632 to W Liu). We acknowledge Professors Quentin Liu for the paclitaxel- and epirubicin-resistant MCF-7 cell lines and Dr. Jue Wang and Dr. Peilin Liao for their technical support. We thank all members of Professor YY Duan's lab for helpful insights and valuable advice.

## Additional information

### Funding

| Funder | Grant reference number | Author |
| --- | --- | --- |
| Guangzhou Municipal Science and Technology Project | 202201020240 | Na Yang |
| Guangdong Medical Research Foundation | A2021112 | Na Yang |
| National Natural Science Foundation of China | 81602434 | Na Yang |
| Research Agreement between South China University of Technology and Guangzhou First People's Hospital | D9194290 | Yuyou Duan |
| National Natural Science Foundation of China | 81972594 | Min Yan |
| Guangzhou Municipal Science and Technology Project | 2023A04J0632 | Wei Liu |

The funders had no role in study design, data collection, and interpretation, or the decision to submit the work for publication.

### Author contributions

Jiamin Lin, Conceptualization, Data curation, Software, Formal analysis, Validation, Investigation, Visualization, Methodology, Writing – original draft, Writing – review and editing; Pingping Zhang, Data curation, Software, Formal analysis, Investigation, Visualization, Methodology, Writing – original draft; Wei Liu, Resources, Data curation, Formal analysis, Funding acquisition, Investigation, Methodology, Writing – original draft, Writing – review and editing, Collected specimen; Guorong Liu, Data curation, Formal analysis, Visualization, Methodology, Collected specimen; Juan Zhang, Data curation, Investigation, Methodology; Min Yan, Yuyou Duan, Resources, Supervision, Funding acquisition, Methodology, Writing – original draft, Project administration, Writing – review and editing; Na Yang, Conceptualization, Resources, Data curation, Formal analysis, Supervision, Funding acquisition, Validation, Visualization, Methodology, Writing – original draft, Project administration, Writing – review and editing

### Author ORCIDs

Jiamin Lin http://orcid.org/0000-0003-1275-7521
Yuyou Duan http://orcid.org/0000-0002-1402-7402
Na Yang http://orcid.org/0000-0003-1383-1615

### Ethics

That informed consent, and consent to publish, was obtained. The specific ethical approval obtained and guidelines that were followed by the Ethics Committee of Guangzhou First People's Hospital. The approval number of the research ethics approval document is K-2021-201.

This study was performed in strict accordance with the recommendations in accordance with Guiding Opinions on the Treatment of Experimental Animals, Guangdong Province Laboratory Animal Management Regulations, and the Laboratory Animal Committee (LAC) of South China University of Technology Policy on the Humane Care and Use of Vertebrate Animals. The Animal Ethics Committee Number is 2020054. All surgery was performed under sodium pentobarbital anesthesia, and every effort was made to minimize suffering.

Reviewer #1 (Public Review): https://doi.org/10.7554/eLife.87510.4.sa1
Reviewer #2 (Public Review): https://doi.org/10.7554/eLife.87510.4.sa2
Reviewer #3 (Public Review): https://doi.org/10.7554/eLife.87510.4.sa3
Author Response https://doi.org/10.7554/eLife.87510.4.sa4

---

## Additional files

### Supplementary files

• Supplementary file 1. The sequences of siRNA target.

• Supplementary file 2. The sequences of gene-specific primers used for quantitative reverse transcriptase polymerase chain reaction (qRT-PCR).

• Supplementary file 3. The control/−287 bp/−965 bp/−1036 bp/−1116 bp and −2000 bp regions and motif1 sequences of primers used for the ACSL4 promoter vector constructs.

• Supplementary file 4. The sequences of gene-specific primers used for chromatin immunoprecipitation (ChIP) assay.

• MDAR checklist

### Data availability

All data generated or analyzed during this study are included in the manuscript and supporting file. The Kaplan–Meier survival analysis used the breast cancer patients in TCGA database (https://www.cancer.gov/ccg/research/genome-sequencing/tcga).

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
