## [Editor Report · eLife assessment]

This study provides a **valuable** finding on the mechanistic connections between epithelial-mesenchymal transition (EMT) and lipid metabolism. The authors identified the ZEB2/ACSL4 axis as a newly discovered metastatic metabolic pathway that promotes both lipogenesis and fatty acid oxidation. The evidence supporting the claims of the authors is **solid**. The work will be of interest to medical biologists working on cancer.

---

## [Referee Report · Reviewer #1 (Public Review)]

In this study, Jiamin Lin et al. investigated the potential positive feedback loop between ZEB2 and ACSL4, which regulates lipid metabolism and breast cancer metastasis. They reported a correlation between high expression of ZEB2 and ACSL4 and poor survival of breast cancer patients, and showed that depletion of ZEB2 or ACSL4 significantly reduced lipid droplets abundance and cell migration in vitro. The authors also claimed that ZEB2 activated ACSL4 expression by directly binding to its promoter, while ACSL4 in turn stabilized ZEB2 by blocking its ubiquitination.

---

## [Referee Report · Reviewer #2 (Public Review)]

In this study, the authors validated a positive feedback loop between ZEB2 and ACSL4 in breast cancer, which regulates lipid metabolism to promote metastasis.

Overall, the study is original, well structured, and easy to read.

---

## [Referee Report · Reviewer #3 (Public Review)]

The manuscript by Lin et al. reveals a novel positive regulatory loop between ZEB2 and ACSL4, which promotes lipid droplets storage to meet the energy needs of breast cancer metastasis.

---

## [Author Response]

The following is the authors’ response to the previous reviews

We appreciate the positive comments from the editors and reviewers. The followings are the point to point responses to the questions and comments of the Reviewers:

**Reviewer #1 (Public Review):**

In this study, Jiamin Lin et al. investigated the potential positive feedback loop between ZEB2 and ACSL4, which regulates lipid metabolism and breast cancer metastasis. They reported a correlation between high expression of ZEB2 and ACSL4 and poor survival of breast cancer patients, and showed that depletion of ZEB2 or ACSL4 significantly reduced lipid droplets abundance and cell migration in vitro. The authors also claimed that ZEB2 activated ACSL4 expression by directly binding to its promoter, while ACSL4 in turn stabilized ZEB2 by blocking its ubiquitination. While the topic is interesting, there are several concerns with the study:

1. My concern regarding the absence of appropriate thresholds or False Discovery Rate (FDR) adjustments for the RNA-seq analysis has not been addressed, leading to incorrect thresholds and erroneous identification of significant signals.

Response: We thank the reviewer for the concern about the RNA-seq analysis. RNA-seq data was analyzed by the Benjamini and Hochberg’s approach for controlling the false discovery rate. The procedure of RNA-seq bioinformatic analysis is as follows:For data analysis, raw data of fastq format were firstly processed through in-house perl scripts. In this step, clean data were obtained by removing reads containing adapter, reads containing N base and low quality reads from raw data. All the downstream analyses were based on the clean data with high quality. Index of the reference genome was built using Hisat2 v2.0.5 and paired-end clean reads were aligned to the reference genome using Hisat2 v2.0.5. FeatureCounts v1.5.0-p3 was used to count the reads numbers mapped to each gene, and then FPKM of each gene was calculated based on the length of the gene and reads count mapped to this gene. Differential expression analysis of two conditions/groups was performed using the DESeq2 R package (1.20.0). The resulting P-values were adjusted using the Benjamini and Hochberg’s approach for controlling the false discovery rate. Genes with an adjusted P-value (<0.05) found by DESeq2 were assigned as differentially expressed.

1. In Figure 3B and C, it appears that the knockdown efficiency of ACSL4 is inadequate in these cells, which contradicts the Western blot results presented in Figure 2F.

Response: We thank the reviewer for the concern. In figure 3B and 3C, we use the shRNA for the knockdown experiment and in Figure 2F we use siRNA for the knockdown experiment, so the efficiency of them were different.

1. Regarding Figure 6, the discovery of consensus binding sequences (CACCT) for ZEB2 alone is insufficient evidence to support the direct binding of ZEB2 to the ACSL4 promoter.

Response: We thank the reviewer for the concern. We performed chromatin immunoprecipitation (ChIP), which examines the direct interaction between DNA and protein, to test if ZEB2 directly binds to the ACSL4 promoter. The results showed that the primer set 1, which covered -184 to -295 of ACSL4 promoter region exhibited apparent ZEB2 binding (Fig. 6F). Moreover, the mutant sequence (AAAA) of ACSL4 promoter showed significant decreased luciferase activity (Fig. 7H). All these evidences suggest that ZEB2 directly bond to the consensus sequence of ACSL4 promoter.

1. For Figure 7E, there are multiple bands present, and it appears that ZEB2-HA has been cropped, which should ideally be presented with unaltered raw data. Please provide the uncropped raw data.

Response: We thank the reviewer for the concern. The raw data of the figure 7E ZEB2-HA is shown in Author response image 1:

**Author response image 1. sa4fig1:** The uncropped raw data of the figure 7E ZEB2-HA.

1. In Figure 7C, the author claimed to have used 293T cells for the ubiquitin assay, which are not breast cancer cells. Moreover, the efficiency of over-expression differs between ZEB2 and ACSL4 in 293T cell lines. Performing the experiment in an unrelated cell line to justify an important interaction is not acceptable.

Response: We thank the reviewer for the concern. We also performed the ubiquitination assay in MDA-MB-231 cells in Fig 7D (Author response image 2), The results confirm that knockdown of ACSL4 obviously enhanced the ubiqutination of ZEB2. We also have performed the IP experiment in MDA-MB-231 cells in Author response image 3 (Fig 7F). The results confirmed the interaction between ZEB2 and ACSL4:

**Author response image 2. sa4fig2:** The ubiquitination assay in MDA-MB-231 cells.

**Author response image 3. sa4fig3:** The IP experiment in MDA-MB-231 cells.

**Reviewer #2 (Public Review):**
In this study, the authors validated a positive feedback loop between ZEB2 and ACSL4 in breast cancer, which regulates lipid metabolism to promote metastasis.Overall, the study is original, well structured, and easy to read.

We appreciate the positive comments from the reviewer.

**Reviewer #3 (Public Review):**
The manuscript by Lin et al. reveals a novel positive regulatory loop between ZEB2 and ACSL4, which promotes lipid droplets storage to meet the energy needs of breast cancer metastasis.

We appreciate the positive comments from the reviewer.

**Reviewer #2 (Recommendations For The Authors):**
I still have some points that should be addressed by the Authors:The interaction between ACSL4 and ZEB2 is still not convincing, due to the cellular localization of ACSL4 and ZEB2 is different. The authors should consider utilizing the Duolink experiment to more accurately determine the interaction location of these two proteins in cells.

Response: We appreciate the reviewer’s suggestion. We performed GST pull-down assay to examine whether ZEB2 and ACSL4 form a complex. GST pull-down assay confirmed the interaction of ZEB2 and ACSL4 (Figure 7—figure supplement 2). We also performed immunofluorescence assay and found that ZEB2 was co-localized with ACSL4 in some certain regions of the cytoplasm in Author response image 5 (Figure 7—figure supplement 3):

**Author response image 4. sa4fig4:** The GST pull-down assay between the ZEB2 and ACSL4.

**Author response image 5. sa4fig5:** The Immunofluorescence co-localization assay of ZEB2 and ACSL4.

In Figure S4, the authors showed both "shACSL4" and "siACSL4", which is a description error.

Response: We appreciate the reviewer to point out the mistake. We have corrected the "siACSL4" into "shACSL4".

**Author response image 6. sa4fig6:** Fluorescence staining of Phalloidin in control or ACSL4 knockdown MDA-MB-231 cells.

**Reviewer #3 (Recommendations For The Authors):**
The manuscript is improved.

We appreciate the positive comments from the reviewer.